# Pro197Thr Substitution in *Ahas* Gene Causing Resistance to Pyroxsulam Herbicide in Rigid Ryegrass (*Lolium Rigidum* Gaud.)

Barbara Kutasy [1] , Zsolt Takács [2], Judit Kovács [3], Verëlindë Bogaj [4], Syafiq A. Razak [4], Géza Hegedűs [5], Kincső Decsi [1], Kinga Székvári [6] and Eszter Virág [5,7,*]

[1] Campus Keszthely, Department of Plant Physiology and Plant Ecology, Hungarian University of Agriculture and Life Sciences Georgikon, 7 Festetics Str., 8360 Keszthely, Hungary; kutasybarbara@gmail.com (B.K.); szaszkone.decsi.kincso@uni-mate.hu (K.D.)

[2] Department of Agricultural Mechanization, University of Pannonia, 16 Deák F. Str., 8360 Keszthely, Hungary; permetezo.takacs@gmail.com

[3] Gregor Mendel Institute of Molecular Plant Biology, Austrian Academy of Sciences, Vienna Biocenter, Dr. Bohr-Gasse 3, 1030 Vienna, Austria; juditkovacsbio@gmail.com

[4] Institute of Plant Protection, University of Pannonia, 16 Deák F. Str., 8360 Keszthely, Hungary; verelindebogaj@gmail.com (V.B.); nanosyafiq92@gmail.com (S.A.R.)

[5] Educomat Ltd., 12/A Iskola Str., 8360 Keszthely, Hungary; hegedus@educomat.hu

[6] Festetics Doctoral School, Hungarian University of Agriculture and Life Sciences, 7 Festetics Str., 8360 Keszthely, Hungary; kinga.szkvr@gmail.com

[7] Department of Molecular Biotechnology and Microbiology, Institute of Biotechnology, Faculty of Science and Technology, University of Debrecen, 1 Egyetem Square, 4032 Debrecen, Hungary

\* Correspondence: eszterandreavirag@gmail.com

**Abstract:** *Lolium rigidum* Gaud. is a cross-pollinated species characterized by high genetic diversity and it was detected as one of the most herbicide resistance-prone weeds, globally. Acetohydroxyacid synthase (AHAS) resistant populations cause significant problems in cereal production; therefore, monitoring the development of AHAS resistance is widely recommended. Using next-generation sequencing (NGS), a de novo transcriptome sequencing dataset was presented to identify the complete open reading frame (ORF) of AHAS enzyme in *L. rigidum* and design markers to amplify fragments consisting of all of the eight resistance-conferring amino acid mutation sites. Pro197Thr, Pro197Ala, Pro197Ser, Pro197Gln, and Trp574Leu amino acid substitutions have been observed in samples. Although the Pro197Thr amino acid substitution was already described in SU and IMI resistant populations, this is the first report to reveal that the Pro197Thr in AHAS enzyme confers a high level of resistance ($ED_{50}$ 3.569) to pyroxsulam herbicide (Triazolopyrimidine).

**Keywords:** target-site resistance; *ahas*; *als*; Triazolopyrimidine herbicide; *Lolium rigidum*

## 1. Introduction

Rigid ryegrass (*Lolium rigidum* Gaud.) is a diploid annual grass species distributed worldwide [1]. *L. rigidum* is one of the most widespread and harmful weeds in the Mediterranean area and it was introduced to North and South America, South Africa, and Australia [2]. Globally, *L. rigidum* occurs as weeds of cereal crops, vineyards, and orchards [3]. Though it was intentionally introduced to south-western Australia as a pasture plant species, it is considered as one of the most problematic weeds [4,5].

*L. rigidum* is one of the most important herbicide-resistant weeds causing economic problems worldwide [6]. Since the 1980s, the development of broad and complex herbicide resistance patterns, such as multiple and cross resistance, has been detected across *L. rigidum* populations [7–9].

However, in weed control the dominant strategy is to use different herbicides that are highly effective on weeds, within these populations, several plants were documented carrying multiple resistance to herbicides including ten different chemical classes [10,11]. Multiple

resistance to PSII, acetyl coenzyme A carboxylase (ACCase) and acetolactate synthase (ALS) inhibitors has been demonstrated in Australian and Spanish populations [5,12,13]. This observation has significant consequences on the treatment management for arable lands, as it indicates that the selection with a single herbicide can cause a complex and highly unpredictable pattern of cross-resistance [14].

An effective herbicide is able to penetrate the plants to their site of action at a lethal dose. Herbicides can inhibit plant specific enzymes on a specific target site, thereby altering plant metabolic mechanisms. If the structure of the target site is changing, then the effect of the herbicide is limited, therefore, plants can develop target-site resistance (TSR) against several herbicides. The mechanism of TSR includes point mutations in a target gene causing amino acid changes and lower herbicide binding capacity to the target enzyme. TSR often involves mutations in genes encoding protein targets of herbicides that affect herbicide binding either at or near the catalytic domains or in regions that affect access to them [15]. The mutant target protein shows increased expression in response to herbicide exposure [16]. Because of the rapid evolution of TSR mutations triggered by genetic and other factors, an increased number of herbicide resistant weed populations can be observed on arable fields [17]. This problem will endanger the sustainability of weed control [18], therefore, it is necessary to detect herbicide resistance in agricultural fields. Determining herbicide resistance in surviving weeds after chemical treatment facilitates the planning and effectiveness of subsequent herbicide treatment. In the next-generation sequencing-targeted amplicon sequencing (NGS-TAS) method, TSR is determined by single nucleotide polymorphism (SNP) genotyping and allows identifying the formed multiple TSR (MTSR) in weed population [19].

A remarkable phenomenon of herbicide resistance in *L. rigidum* cross-pollinated weed species, is the high rate of occurrence of cross-resistance that has been confirmed globally. For instance, resistance to AHAS (acetohydroxyacid synthase, EC 4.1.3.18, also known as ALS) inhibitors has been found in Australia, Chile, France, Greece, Israel, and South Africa [20]. Moreover, triasulfuron-resistant populations were found to be resistant also to imidazolinone herbicides [21]. As a consequence of amino acid substitutions, resistant plants were detected in the genetically diverse and resistance-prone *L. rigidum* genus. Examining a single population, 1–6 different mutations were found in the *ahas* gene sequence causing herbicide resistance [22]. So far, 13 Sites of Action were detected in resistant populations in the case of 35 herbicides, therefore, *L. rigidum* was ranked as one of the most herbicide resistance-prone weeds [20].

Resistance to AHAS targeting herbicides is the most common problem among weed species. Most weed species are more resistant to AHAS-inhibiting herbicides than to any other herbicides [8,23]. AHAS catalyzes the first common step in the pathway for synthesis of the branched-chain amino acids such as leucine, isoleucine, and valine in plants. This herbicide group consists of five different chemical families: sulfonylureas (SU), imidazolinones (IMI), sulfonylaminocarbonyl-triazolinones (SCT), pyrimidinylthiobenzoates (PTB), and triazolopyrimidines (TP). AHAS inhibitor herbicides have been widely used as a result of their effectiveness against several weeds at low doses with low mammalian toxicity and their selectivity for major crops [24,25]. Pyroxsulam (molecular formula: $C_{14}H_{13}F_3N_6O_5S$) (IUPAC name: N-(5,7-dimethoxy[1,2,4] triazolo[1,5-a]pyrimidin-2-yl)-2-methoxy-4- (trifluoromethyl) pyridine-3-sulfonamide) is a member of the TP chemical family. Pyroxsulam herbicide inhibits the production of the AHAS enzyme in plants. Therefore, it can control annual grasses and broadleaf weeds in cereal crops, corn, soybean, sunflower, cotton, tree, and vine crops.

A few point mutations may occur in the sequence of a target enzyme gene, evolving resistance to herbicides. In plants, point mutations causing amino acid substitutions at eight different codons of the *ahas* gene, have been confirmed to cause resistance in field-evolved resistant weed species [17,26]. So far in AHAS enzyme, two amino acid substitutions at the position of Pro197 and Trp574 (according to *Arabidopsis thaliana* numbering) have been reported in *L. rigidum* samples as a consequence of SU and IMI. Considering

AHAS enzyme, six resistance-endowing amino acid substitutions were identified in one *L. rigidum* population: Pro197Ala, Pro197Arg, Pro197Gln, Pro197Leu, Pro197Ser, and Trp574Leu [20,22,27]. Furthermore, Ala122Gly, Pro197Thr, Ala205, Asp376Asn, Asp376Glu amino acid changes were observed in Greek and Italian populations after SU and IMI treatment [28,29]. Although AHAS has a crucial role in the development of herbicide resistance, interestingly, its whole gene sequence is still unknown. It would be essential to identify it to be able to discover new amino acid substitutions in order to fully understand its role in herbicide resistance.

*L. rigidum*, as a highly susceptible weed, poses a serious economic threat to intensive cropping systems. The use of herbicides is a dominant part of integrated weed management practices, so the knowledge of developed herbicide resistance is important (Beckie 2006). Our aim was to determine the degree of AHAS resistance and the type of resistance (TSR or NTSR) in the suspected Tunisian population of *L. rigidum*. The site resistance can be confirmed by examining mutations that cause known resistance.

In this study, the complete AHAS enzyme mRNA sequence in sensitive *L. rigidum* was identified using Illumina de novo sequencing technology, which sequence information was gap-filling information in this area. After pyroxsulam treatment resistant populations were produced in order to analyze the mutations in the *ahas* gene coding sequences. Based on Illumina data, point mutations were identified using Sanger sequencing (Figure 1).

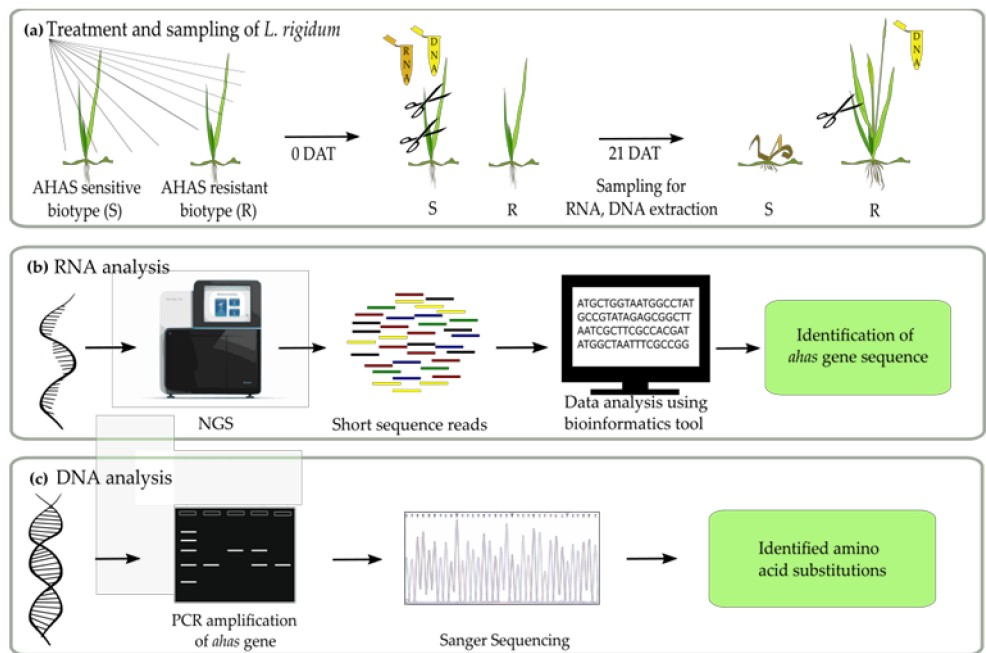

**Figure 1.** The investigation of the *ahas* gene in *L. rigidum* population. (**a**) Pyroxsulam (TP herbicide) treatment and collection of leaf tissue for RNA and DNA isolation at 0 DAT and for DNA isolation at 21 DAT. (**b**) Process of RNA analysis. After RNA extraction, library preparation was prepared for Illumina NextSeq500 paired-end sequencing. The quality of the raw reads was checked, and de novo assembly of transcripts was performed using Trinity software and the ORF of the *ahas* gene was identified. (**c**) DNA analysis. After DNA extraction, gene fragments of the *ahas* gene were amplified by PCR and their nucleotide sequence was identified by Sanger sequencing. As a result, amino acid substitutions were identified.

## 2. Materials and Methods

### 2.1. Plant Materials and Growth Conditions

Seeds of 50 putative pyroxsulam-resistant *L. rigidum* mature plants were collected from a wheat field beside Ben Arous city, in north-eastern Tunisia where pyroxsulam (Pallas^TM 45 OD, Dow AgroSciences) was used at a normal field dose of 18.4 g ai/ha. Immediately, after collection of all seeds, they were air-dried and stored at 4 °C until used in consecutive

experiments. Sensitive seeds of *L. rigidum* that have never been treated with herbicides were obtained from Seed Bank of Hungarian University of Agriculture and Life Sciences Georgikon Campus, Institute of Plant Protection. Seeds were germinated in Petri dishes at 30 °C/20 °C 16 h photoperiod and were planted in pots and were stored in a greenhouse.

### 2.2. De Novo Transcriptome Sequencing and Assembly

Leaves of five herbicide-susceptible samples were collected and frozen in liquid nitrogen immediately after collection and were stored at −80 °C. The total RNA from plant tissues were isolated using TaKaRa Plant RNA Extraction Kit following the manufacturer's instruction (Takara Bio Inc.; Shiga, Japan). The purity and concentration of the RNA sample were assessed on the Agilent TapeStation system (Agilent Technologies Inc.; Santa Clara, CA, USA) and Qubit 4 Fluorometer (Thermo Fisher Scientific; Waltham, MA, USA) (Table S1, Figure S1). As starting material, 2.0 μg of total RNA was used from which cDNA synthesis and library construction were performed. TruSeq™ RNA sample preparation kit (Low-Throughput protocol) was used to enrich the mRNA, synthesize cDNA, and prepare the library for Illumina NextSeq paired-end sequencing (Figure S2). The quality of the raw (2 × 80 base long) reads was checked by using FastQC quality control software [30]. Quality trimming at a Phred score ≥30 was performed by using self-developed software [31]. De novo assembly of transcripts was performed using Trinity [32] with 25 k-mer size. Assembled contigs were used as *L. rigidum* reference transcriptome. The transcript abundances were estimated using the RSEMprogram (http://deweylab.github.io/RSEM/ accessed on 10 June 2021) with the scripts provided in the Trinity package. Raw reads and assembled transcript dataset were deposited in the NCBI Sequence Read Archive (SRA: SRR8469510) and Transcriptome Shotgun Assembly (TSA: GHHB00000000.1; BioProject PRJNA512627).

### 2.3. Identification of Ahas mRNA in L. Rigidum

*Ahas* mRNA of *Lolium multiflorum* (GenBank: AF310684.2) was used as a reference. To dynamically translate all reading frames, Blastn algorithm was used [33]. The mRNA sequences were identified from BLAST and aligned into a single gene by using ClustalW [34]. Open reading frames (ORF) were identified using NCBI ORF finder (https://www.ncbi.nlm.nih.gov/orffinder/ accessed on 10 June 2021). The investigated sequence of *L. rigidum* was deposited in the NCBI GenBank MK492446.

### 2.4. Dose-Response Analysis

Twenty putative pyroxsulam-resistant plants at the three-leaf stage in each pot were treated with pyroxsulam herbicide as commercial herbicide formulation PallasTM 45 OD, using a plot sprayer delivering 250 L ha$^{-1}$, at a pressure of 300 kPa and a speed of 1.2 ms$^{-1}$, with a boom equipped multirange flat fan hydraulic spray nozzle (TeeJetR© XR11002). Relative biomass reduction bioassay was performed using six application rates: 0, 4.6, 9.2, 18.4, 36.8, 73.6, and 147.2 (18.4 g ai/ha as the recommended field dose). The treatment was applied on sensitive samples, as control and on resistant samples. Leaves of sensitive samples were collected at 0 day after treatment (DAT) for DNA extraction. The pots were placed in a greenhouse and survivors were assessed at 21 DAT. The herbicide treatment was applied in three technical replicates with a complete randomized block. The dose-response experiment was repeated once more. The nonlinear regression analysis was done using the "drc" package in R v4.0.4 (https://cran.r-project.org/web/packages/drc/drc.pdf accessed on 10 June 2021) (Figure S3). According to the biomass weight of the resistant population, effective dose (ED$_{50}$) was estimated using the four-parameter logistic Equation (1) [35,36]:

$$y = C + (D - C)/[1 + (x/ED_{50})^{b}] \tag{1}$$

ED$_{50}$, the effective dose causing 50% reduction in biomass, indicates the level of resistance (R/S ratios).

### 2.5. Ahas Gene of L. Rigidum Amplification and Sequencing

Total DNA was extracted from leaf tissues of 50–50 putative pyroxsulam-resistant and sensitive samples [37]. Three primer pairs were designed using the software Primer3Plus to amplify the earlier described resistance-conferring mutation sites based on the *ahas* mRNA in *L. rigidum* (Table S1). The PCR was performed in a final volume of 15 µL (Thermo Fisher Scientific; Waltham, MA, USA). Amplification was carried out in an Eppendorf Mastercycler ep384 (Eppendorf AG; Hamburg, Germany). PCR products were controlled with Agilent 2100 Bioanalyzer (Agilent Technologies Inc.; Santa Clara, CA, USA) using Agilent DNA 1000 Kit and were purified using NucleoSpin Gel and PCR Clean-up System (Macharey-Nagel GmbH &Co; Düren, Germany). The pGEM-T Vector System II (Promega; Madison, WI, USA) was used for cloning according to the manufacturer's instructions and fragments were sequenced ABI 3130x1 Genetic Analyzer (Applied Biosystems; Carlsbad, CA, USA). This 16-capillary array genetic analyzer was used with 50 cm capillary and 3130 POP-7 separation polymers.

### 3. Results

In this study, an RNA-seq approach is presented to identify complete coding sequences (cds) of the *ahas* gene in *L. rigidum* and mutation points involved resistance to TP herbicide, pyroxsulam. A novel amino acid substitution was identified among the earlier described resistance-conferring mutation sites using Sanger sequencing.

### 3.1. Transcriptome Reconstruction and Identification of Ahas Cds

The *L. rigidum* mRNA library was sequenced using NextSeq500 platform. After quality filtering, 18,636,868 cleaned reads were obtained from the total raw reads of 18,642,689 ($2 \times 80$ bp paired-end) showing Phred score 34 and GC% 55 on average. Based on FastQC report, sequences represented more than 0.1% of the total and low-quality bases (corresponding to a 0.1% sequencing error rate) were removed. After transcriptome assembly by Trinity, de novo transcript dataset resulted in 74,279 contigs with lengths from 201 to 15,531bp (GC% 52.5). This transcript dataset was used to identify mRNA of *ahas* gene in *L. rigidum*. The DN19950_c0_g2_i3 contig was used to determine the ORF of *ahas* gene. The similarity between *L. multiflorum* as the reference and the identified cds in *L. rigidum* was 98%. The determined *ahas* mRNA sequence was 1929bp long.

### 3.2. Dose–Response Experiment

The population of *L. rigidum* contains different amino acid substitutions. Each sample contains one of the amino acid substitutions of Pro197Thr, Pro197Ala, Pro197Ser, Pro197Gln, or Trp574Leu. The resistance patterns of the combined substitution population were characterized. The estimated $ED_{50}$ is 3.569 that indicates the 50% survival of the resistance population to pyroxsulam that showed Pro197 or Trp574 substitution (Figure 2). Following a pyroxsulam field application rate of 18.4 g ha$^{-1}$, *L. rigidum* putative pyroxsulam-resistant population survived with 98.9% growth reduction.

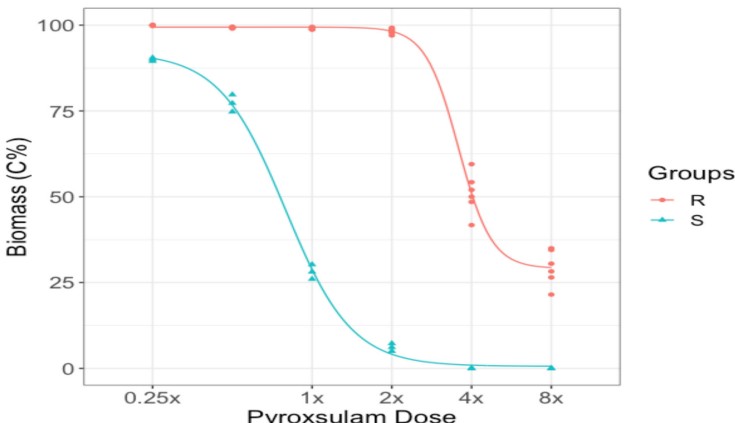

**Figure 2.** Effect of pyroxsulam herbicides on biomass reduction of *L. rigidum* population. Dose-response curve of biomass weight after pyroxsulam treatment. Data points indicate mean values of 20 independent replicate experiments (average n = 120 plants per dose). Red line represents a 4-parameter logistic regression of pyroxsulam-resistant population. Green line represents sensitive samples.

*3.3. Determination of Mutation Points of Ahas Gene in L. Rigidum*

Three parts of the *ahas* gene were amplified and sequenced by Sanger sequencing. The primer pairs covered all of eight resistance-conferring amino acid mutation sites at three different regions: 418, 259, and 399 bp (Figure S4, Table S2). The *Ahas* gene fragments were amplified by PCR from 50–50 resistant and sensitive samples and sequenced by the Sanger method (Figure S5). Results of sequence alignment analysis showed single nucleotide substitutions in Pro197 and Trp574 positions in resistant samples (Figure 3). Wild type of *L. rigidum* samples showed no mutations at the eight known nucleotide sequences of the *ahas* gene that correspond to the resistance.

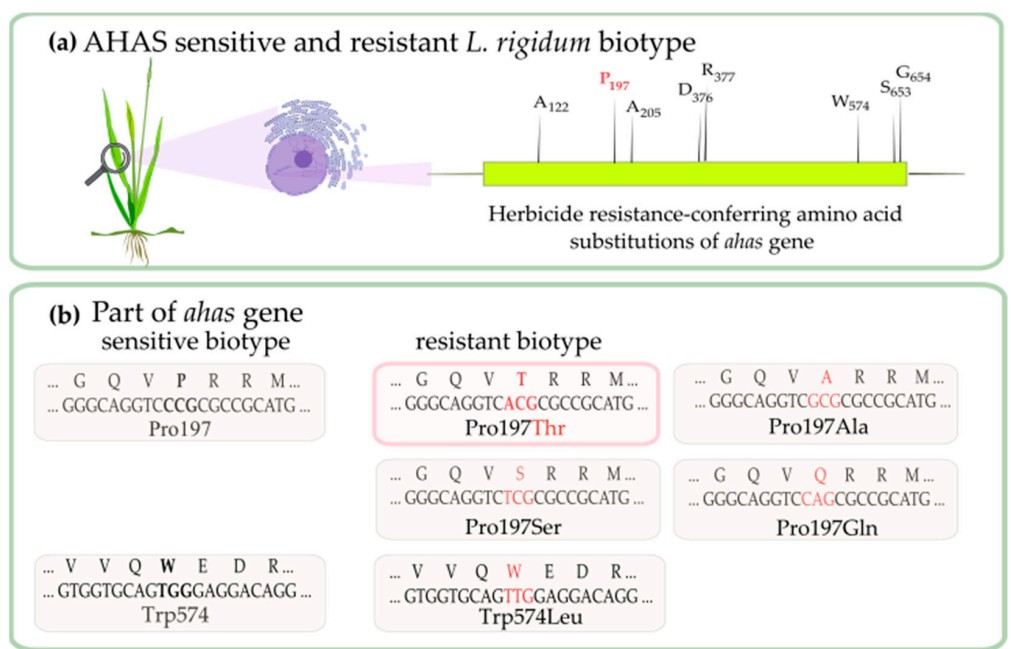

**Figure 3.** Analysis of the *ahas* gene nucleotide sequence localized in the nucleus in AHAS sensitive and resistant *L. rigidum* biotype. (**a**) The eight known amino acid position in the *ahas* gene responsible for resistance in *L. rigidum*. (**b**) Collection of the amino acid substitutions (red) at the 197 and 574 positions in AHAS sensitive and resistant biotypes. Box highlighted in red indicates the newly discovered amino acid substitution (Pro197Thr).

Sequence alignment analysis showed a single nucleotide change of CCG to ACG at the 197 positions that resulted in a Pro/Thr substitution in the resistant samples (Figures S6 and S7). This is the first Pro197Thr substitution in an *ahas* resistant *L. rigidum* population that was reported after triazolopyrimidines herbicide treatment. Moreover, amino acid mutations were detected at Pro197Ala, Pro197Ser, Pro197Gln, and Trp574Leu (Figure S8) in resistant samples as previously reported [22]. A nucleotide substitution (GAT/GAC) was observed at the Asp376 amino acid position that did not evolve amino acid substitution (Figure S9).

## 4. Discussion

In arable lands, chemical weed control is one of the most common and unavoidable methods of controlling weeds, which means high selection pressure for resistant weeds. The presence of high levels of herbicide resistance risks the sustainability of current weed control strategies, the integrated weed control [38,39]. In this study, Illumina RNA-sequencing and the "gold standard" Sanger sequencing were used to investigate the sequence of the ORF *ahas* gene in *L. rigidum*. Rigid ryegrass was noticed as a highly resistance-prone weed species and numerous populations were described as resistant to AHAS inhibitors, including SU and IMI. Populations containing Pro197 Ala122, Ala205, Asp376, and Leu 574 amino acid mutations showed high resistance (RI from 8 to 70) after SU treatment. Types of mutant ALS alleles, such as Pro197Ala, Pro197Gln, Pro197Leu, Pro197Ser, and Pro197Thr were observed in these samples [28]. The same biotypes were detected in our samples after pyroxsulam treatment. This is the first report that reveals that the Pro197Thr amino acid change in the AHAS enzyme can confer a high level of resistance (ED$_{50}$ is 3.569) to pyroxsulam (TP) herbicide in *L. rigidum*. The different AHAS-mutations confer different resistance against the AHAS inhibitor herbicide chemical families [20]. It has been already shown that several weed species carrying the Pro197 and Thr574 amino acid substitutions developed high levels of resistance after TP herbicide treatment. In this study, the investigated *L. rigidum* population that contained amino acid substitutions at the 197 and 574 positions, also showed high resistance against the pyroxsulam herbicide. So far, the Pro197Thr resistance-endowing mutation position has been reported in four weed species after TP treatment [20]. Substitutions at Pro197Thr are known to cause a high level of resistance to TP in *Raphanus raphanistrum* [40], in *Kochia scoparia* [41] and moderated resistance to TP in *Apera spica-venti* [26] and *Alopecurus aequalis* [42]. In this study, fifty *L. rigidum* resistant plants were examined and every individual carried only one mutation point out of the eight in the *ahas* gene at the known positions of resistance-endowing amino acid substitutions. Single Pro197Thr substitution was observed in 16 samples (Table S3). As the pyroxsulam resistant biotypes contained different single amino acid substitution in the *ahas* gene, the statistical analysis describes the mixed population. To obtain more information about the Pro197Thr amino acid substitution in the *ahas* gene, larger populations need to be examined. Moreover, the Thr197Pro mutation was described, after SU and IMI treatment, as the same mutation that occurs after pyroxsulam treatment; therefore, presence of cross-resistance is presumed. The existence of cross-resistance in this allelic variant should be demonstrated by treatments with other herbicides belonging to other AHAS inhibitor families (e.g., SU, IMI). Additional herbicides that demonstrate the presence or absence of cross-resistance will be included in our further experiments. Indeed, the detected mutation would provide important information about possible cross-resistance and can help to plan treatment with AHAS inhibitors.

## 5. Conclusions

Due to the rapid evolution of herbicide resistance, it is necessary to constantly monitor resistant populations on the arable fields in order to maintain weed control considering environmental protection and sustainable agriculture. The pyroxsulam-resistant *L. rigidum* population showed a TSR mutation detected at positions Pro197 and Leu574. The knowledge of resistance-inducing effects of compounds containing AHAS, including

pyroxsulam-triggered TSR, would help to develop diagnostic methods, such as NGS-TAS. By diagnosing the presence of TSR and MTSR, chemical weed control would become more predictable, and it will guide farmers on how to plan integrated and resistant weed management strategies that are less harmful to the environment.

**Supplementary Materials:** The following are available online at https://www.mdpi.com/article/10.3390/su13126648/s1, Figures S1–S9: Data of RNA sample, Statistic data of *drc* analysis, Primers, Gel pictures and Electropherogram of Sanger sequencing. Tables S1–S3: Data of RNA sample, Primers, Data of amino acid substitutions.

**Author Contributions:** Conceptualization, B.K. and Z.T.; methodology, B.K., Z.T., V.B., S.A.R., K.D. and K.S.; software, G.H.; validation, B.K., E.V. and G.H.; formal analysis, B.K.; investigation, B.K. and E.V.; resources B.K.; data curation, E.V.; writing—original draft preparation, B.K., J.K. and E.V.; writing—review and editing, J.K. and E.V.; visualization B.K. and J.K.; supervision, E.V. and J.K.; project administration, B.K., E.V.; funding acquisition, B.K. and K.D. All authors have read and agreed to the published version of the manuscript.

**Funding:** This research received no external funding. The work was supported by Hungarian University of Agriculture and Life Sciences, Institute of Agronomy, Department of Plant Physiology and Plant Ecology, Gödöllő, 1 Páter Károly str., 2100 Gödöllő, Hungary.

**Institutional Review Board Statement:** Not applicable.

**Informed Consent Statement:** Not applicable.

**Data Availability Statement:** Campus Keszthely, Department of Plant Physiology and Plant Ecology, Institute of Agronomy, Hungarian University of Agriculture and Life Sciences Georgikon, 7 Festetics str., 8360 Keszthely, Hungary.

**Acknowledgments:** We express our thanks to Peter Nagy, DowDuPont Inc. Mougins, France and Janos Taller, Hungarian University of Agriculture and Life Sciences (MATE) to release available the resistant *L. rigidum* seeds (biological samples) to our experiments. Moreover, we express our thanks to László Kocsis, Government Office of Fejér County, Velence to perform the chemical spraying. We are grateful to the editor and to two anonymous reviewers for valuable comments that have helped to improve the manuscript.

**Conflicts of Interest:** The authors declare no conflict of interest. The funders had no role in the design of the study; in the collection, analyses, or interpretation of data; in the writing of the manuscript, or in the decision to publish the results.

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
