# Peer review of "Pro197Thr Substitution in Ahas Gene Causing Resistance to Pyroxsulam Herbicide in Rigid Ryegrass (Lolium Rigidum Gaud.)"

_sustainability, doi:10.3390/su13126648_

Round 1
Reviewer 1 Report
The manuscript includes a high quantity of data produced with the NGS, maybe too many to justify the poor results obtained. The authors reported the orf of the AHAS gene which has been already widely studied in Lolium species, and many sequences are available on genbank database. Furthermore, they stated that the mutation Pro197Thr was firstly reported in Lolium in their research but this is not true, see Scarabel et al. 2020, in Frontiers in Plant Science, doi: 10.3389/fpls.2020.608845.
Author Response
Dear Reviewer #1,
Thank you for your valuable review and comments.
The English language of the manuscript was corrected.
We supplemented the manuscript with the latest research findings. Another 6 publications have been included in the manuscript, which contain the latest relevant findings related to the topic. Please see the following publications:
Scarabel L, Panozzo S, Loddo D, Mathiassen SK, Kristensen M, Kudsk P, Gitsopoulos T, Travlos I, Tani E and Chachalis D (2020) Diversified resistance mechanisms in multi-resistant Lolium spp. in three European countries. Frontiers in plant science 11. doi: 10.3389/fpls.2020.608845
Anthimidou E, Ntoanidou S, Madesis P and Eleftherohorinos I (2020) Mechanisms of Lolium rigidum multiple resistance to ALS-and ACCase-inhibiting herbicides and their impact on plant fitness. Pesticide biochemistry and physiology 164:65-72. doi: 10.1016/j.pestbp.2019.12.010
Gaines TA, Duke SO, Morran S, Rigon CA, Tranel PJ, Küpper A and Dayan FE (2020) Mechanisms of evolved herbicide resistance. Journal of Biological Chemistry 295:10307-10330.
Torra J, Montull JM, Taberner A, Onkokesung N, Boonham N and Edwards R (2021) Target-Site and Non-target-Site Resistance Mechanisms Confer Multiple and Cross-Resistance to ALS and ACCase Inhibiting Herbicides in Lolium rigidum From Spain. Frontiers in plant science 12:88. doi: 10.3389/fpls.2021.625138
Owen MJ, Martinez NJ and Powles SB (2014) Multiple herbicide‐resistant Lolium rigidum (annual ryegrass) now dominates across the Western Australian grain belt. Weed Research 54:314-324. doi: 0.1111/wre.12068
Neve P and Powles S (2005) High survival frequencies at low herbicide use rates in populations of Lolium rigidum result in rapid evolution of herbicide resistance. Heredity 95:485-492. doi: 10.1038/sj.hdy.6800751
Comment 1: The manuscript includes a high quantity of data produced with the NGS, maybe too many to justify the poor results obtained.
Answer 1: Our research group is working on different herbicide target genes of Lolium rigidum. This study present only a part of Lolium rigidum herbicide target studies. In this study we focuses on the ahas gene exclusively.
In the present study we use a large NGS dataset to identify the whole ORF sequence of L. rigidum ahas gene. It may be seem too many for this information, however generation of NGS-based transcriptome sequences was required for subsequent studies as well. We think, the dataset is valuable because the SRA and TSA database is available to anyone in the NCBI. Even though results are not a large they are specific presenting extensive studies, and contribute to the supplementation of the results so far.
Comment 2: The authors reported the orf of the AHAS gene which has been already widely studied in Lolium species, and many sequences are available on genbank database.
Answer 2: Indeed, the ahas gene ORF has been extensively studied in Lolium rigidum, with 21 sequences in the NCBI database, but each are only partial sequences. We identified the complete ORF sequence Lolium rigidum ahas, which complements the previous knowledge.
Comment 3: Furthermore, they stated that the mutation Pro197Thr was firstly reported in Lolium in their research but this is not true, see Scarabel et al. 2020, in Frontiers in Plant Science, doi: 10.3389/fpls.2020.608845.
Answer 3: We would like to report in this study that the Pro197Thr mutation in Lolium rigidum was observed after Triazolopyrimidine (TP) herbicide treatment that is a new observation. Pro197Thr after sulphonylureas (SU) treatment has been reported in the literature, as in the indicated article (iodosufuron-methyl-sodium + mesosulfuron-methyl) and after treatment with an active substance belonging to the IMI chemical family. This has been clarified:
“This is the first report revealing that the Pro197Thr amino acid substitution in AHAS enzyme confers a high level of resistance to Triazolopyrimidine (TP) herbicide.
Rigid ryegrass was noticed as a highly resistance-prone weed species and numerous populations were described as resistant to AHAS inhibitors including SU and IMI. This is the first report which reveals that the Pro197Thr amino acid change in the AHAS enzyme can confer a high level of resistance to TP herbicide”.
We hope that our answers are satisfactory and the corrected manuscript is acceptable for publication.
Sincerely,
Eszter Virág
Corresponding author
Reviewer 2 Report
The manuscript is interesting. In my opinion, it should be expanded with a substantive summary.
Author Response
Dear Reviewer #2,
Thank you for your valuable review and comments.
We have prepared a Conclusion section that summarizes the results and outlines future goals.
We hope that the corrected manuscript is acceptable for publication.
Sincerely,
Eszter Virág
Corresponding author
Reviewer 3 Report
The manuscript entitled "Novel amino acid substitution in ahas gene causing resistance to Triazolopyrimidine herbicide in rigid ryegrass (Lolium rigidum Gaud.)" aims to evaluate the Pro197Thr amino acid substitution in AHAS enzyme confers a high level of resistance to Triazolopyrimidine (TP) herbicide. I consider this study interesting.
Nonetheless, I have some comments below :
- The abstract should be well written with clear justification and proper explanation.
- Proper grammatical error checking.
- Make an absolute “Conclusion” section.
Author Response
Dear Reviewer 3,
Thank you for your valuable review and comments.
Please find our answers to your comments below.
Comment: The manuscript entitled "Novel amino acid substitution in ahas gene causing resistance to Triazolopyrimidine herbicide in rigid ryegrass (Lolium rigidum Gaud.)" aims to evaluate the Pro197Thr amino acid substitution in AHAS enzyme confers a high level of resistance to Triazolopyrimidine (TP) herbicide. I consider this study interesting.
Nonetheless, I have some comments below:
- The abstract should be well written with clear justification and proper explanation.
- Proper grammatical error checking.
- Make an absolute “Conclusion” section.
Answer: The abstract was supplemented with key informations from the results section which are: plant location, drug name, degree of resistance (ED50), mutations that occurred, and an explanation of the severity of the problem.
English grammatical errors have been corrected.
We have prepared a Conclusion section that summarizes the results and outlines future goals.
We hope that our answers are satisfactory and the corrected manuscript is acceptable for publication.
Sincerely,
Eszter Virág
Corresponding author
Reviewer 4 Report
Dear Authors,
The present study presented a novel amino acid substitution in ahas gene that confers high level of resistance to pyroxsulam. Regarding manuscript structure and style, the study is well-organized, with an adequate structure. The scientific originality is acceptable, but the experimental methods need some improvements. The level of English is adequate; however it is suggested to be checked by a native English speaker. In general, tables and figures are easy to understand. Below, you can find detailed comments and suggestions on the manuscript that will help to improve the quality of the manuscript.
Abstract. Please add any necessary key information from the results section and their main interpretations. Please also avoid using first plural (e.g. line 25: …we presented…) throughout the manuscript.
Title. Maybe it the Pro197Thr substitution as well as the name of the triazolopyrimidine herbicide (pyroxsulam) could be included in the title in order to be more concise and specific.
Introduction. The last paragraph could be rephrased in a way that would present firstly what was the the main aim of the work and after that highlight the main conclusions.
Materials and Methods. It is important to be clarified how the pyroxsulam-resistant L. rigidum plants were characterized as resistant. Were they characterized as resistant due to their ability to survive the recommended dose? In that case, before a screening test or a dose-response experiment it might be more accurate to characterize them as putative resistant plants.
Moreover, in line 132 on page 4 it is mentioned that “Leaves of five herbicide-susceptible biotypes were collected…”. Similarly, in lines 165 and 166 on page 4 “sensitive” and “resistant” biotypes were mentioned. As a biotype is considered as a group of plants within a species that has biological traits that aren’t common to the population as a whole. Thus, the 50 pyroxsulam resistant L. rigidum plants that were selected from the wheat filed beside Ben Arous city (line 121-122, page 3) are considered as a putative resistant biotype. Similarly, if it is well understood the sensitive seeds of L. rigidum that have never been treated with herbicides (lines 125-126, page 3) are considered as a susceptible biotype. It is important to clarify how many resistant and sensitive biotypes you have throughout the manuscript and be consistent with their characterization/naming (e.g plants, biotypes, samples).
Results. Regarding dose-response experiment, in materials and methods it is mentioned that for dose response experiment both sensitive and resistant biotypes were used. However, in Figure 2 there is only one common curve and in the caption it is mentioned that this curve refers to L. rigidum population. In such an experiment, it would be crucial to have the curves of both sensitive and resistant biotypes.
Discussion. This section needs to be reinforced with more relevant study data for target site resistance in L. rigidum and their interpretations and also add some key references (for instance Heap, I. The International Herbicide-Resistant Weed Database. Available www.weedscience.org). Finally, future research directions could also be mentioned.
Kind regards,
The Reviewer
Author Response
Dear Reviewer 4,
Please find the answers to your comments in the attachment.
Sincerely, Eszter Virag

Round 2
Reviewer 1 Report
Dear authors,
thanks for the replies to my doubts. Anyway, I am still dubious.
The effort made to improve the paper and reinforce the conclusions and innovations found with this research is appreciable, but I am still not convinced. Furthermore, the authors have put a lot of emphasis on the new allelic variant that gives resistance to TP, but 1) only pyroxulam has been tested (resistance to a single active ingredient of a family does not mean resistance to the whole family of herbicides, see Yu et al., 2012 https://doi.org/10.1111/j.1365-3180.2012.00902.x) and 2) there is no data that says that this allelic variant gives (or does not) cross-resistance to other family of ALS inhibitors. What I mean is that if this allelic variant gives resistance to both SU and IMI in Lolium rigidum (as evidenced by other research cited), I do not see how resistance to one TP would be an added value for the scientific community or for the management of this biotype.
Anyway, I decided to accept the paper withy minor revision, because an improvement has been done. Please, check again the entire manuscript to be sure that no general statements are included when only data on one resistant population resistant to one herbicide has been tested.
Author Response
Dear Reviewer,
Thank you for your valuable review and comments.
The English language of the manuscript was corrected, the minor spell check was done.
comment 1
only pyroxulam has been tested (resistance to a single active ingredient of a family does not mean resistance to the whole family of herbicides, see Yu et al., 2012 https://doi.org/10.1111/j.1365-3180.2012.00902.x)
answer 1
Indeed, the result of a specifically performed experiment is presented in this manuscript, so we also see the need to continue and extend the studies. In the Abstact, Discussion, and Conclusion sections we improved the wording of the results. The generally described name Triazolopyrimidine has been improved to pyroxsulam herbicide. This is how we intend to avoid erroneous conclusions.
comment 2
there is no data that says that this allelic variant gives (or does not) cross-resistance to other family of ALS inhibitors. What I mean is that if this allelic variant gives resistance to both SU and IMI in Lolium rigidum (as evidenced by other research cited), I do not see how resistance to one TP would be an added value for the scientific community or for the management of this biotype.
answer 2
In the manuscript, we highlighted that the existence of cross-resistance in this allelic variant should be demonstrated by treatments with other herbicides belonging to other AHAS inhibitor families (e.g. SU, IMI). Additional herbicides that demonstrate the presence or absence of cross-resistance will be included in our experiments. Indeed, the detected mutation would provide important information about possible cross-resistance and can help to plan treatment with AHAS inhibitors.
We hope that our answers are satisfactory and the corrected manuscript is acceptable for publication.
Sincerely,
Eszter Virág
Corresponding author
09 Jun 2021